# Evaluation and Comparison of Multi-Omics Data Integration Methods for Subtyping of Cutaneous Melanoma

**DOI:** 10.3390/biomedicines10123240

**Published:** 2022-12-13

**Authors:** Adriana Amaro, Max Pfeffer, Ulrich Pfeffer, Francesco Reggiani

**Affiliations:** 1IRCCS Ospedale Policlinico San Martino, 16132 Genova, Italy; 2Faculty of Mathematics, Technical University of Chemnitz, 09111 Chemnitz, Germany

**Keywords:** multi-domain data, cancer genomics, data fusion, tumor classification

## Abstract

There is a growing number of multi-domain genomic datasets for human tumors. Multi-domain data are usually interpreted after separately analyzing single-domain data and integrating the results post hoc. Data fusion techniques allow for the real integration of multi-domain data to ideally improve the tumor classification results for the prognosis and prediction of response to therapy. We have previously described the joint singular value decomposition (jSVD) technique as a means of data fusion. Here, we report on the development of these methods in open source code based on R and Python and on the application of these data fusion methods. The Cancer Genome Atlas (TCGA) Skin Cutaneous Melanoma (SKCM) dataset was used as a benchmark to evaluate the potential of the data fusion approaches to improve molecular classification of cancers in a clinically relevant manner. Our data show that the data fusion approach does not generate classification results superior to those obtained using single-domain data. Data from different domains are not entirely independent from each other, and molecular classes are characterized by features that penetrate different domains. Data fusion techniques might be better suited for response prediction, where they could contribute to the identification of predictive features in a domain-independent manner to be used as biomarkers.

## 1. Introduction

Next-generation sequencing (NGS) technologies have opened the way for the production of huge amounts of genomic data, which can potentially shed light on the complex biological mechanisms that connect genotypes to phenotypes [1]. In the last decade, tumor data from different domains, such as DNA methylation, gene expression, copy number alteration (CNA), and somatic mutation data, have been made publicly available in repositories such as the Cancer Genome Atlas, which annotates published data with relevant clinical and molecular information [2].

Specific clinically relevant phenotypes are regulated by multiple genomic domains. Focusing an analysis on a single domain might reduce the potential to correctly predict the development of a disease and the response to therapy. Mutations of the breast cancer genes 1 and 2 (BRCA1, 2), for example, determine the development of homologous recombination deficiency (HRD) in ovarian cancer and make tumors more sensitive to PARP inhibitors [3,4]. The same effect could, however, also be induced via the methylation of the wild-type genes. In this case, the use of genetic variant information only would potentially exclude patients from beneficial treatments [3,5].

Data fusion (DF) techniques have emerged as a way to merge the information contained in different domains, with the aim of performing feature extraction or patient or sample cluster analyses while considering multiple sources of genomic information at the same time [6]. Data fusion is distinct from more traditional data analysis methods due to the generation of a single multi-domain dataset that can be used for classification. Data merging, on the contrary, consists of the generation of a single dataset using several datasets of the same genomic domain, while data integration refers to the identification of matches between several single-domain datasets that are analyzed separately, such as copy number and gene expression data or gene expression and DNA methylation data analyzed in a gene-centric manner [6,7].

Similarity network fusion (SNF) is a commonly used, heuristic DF clustering method that simply involves computing a sample correlation network for each domain that is then merged into a meta network based on the similarity of the single-domain networks. SNF allows for sample clustering, which can yield clusters with enriched, clinically relevant features [8,9]. Data fusion methods have been applied to full and reduced datasets [9,10]. In the second case, only a subset of genes is selected via unsupervised data filtering to be considered for further analysis. The filtering of biological datasets generally relies on the coefficient of variation (CV as the standard deviation divided by the mean) [9,10] or median absolute deviation filtering (MAD as the median of the absolute difference between each element and the median) [11].

After preprocessing, data fusion methods are applied and class discovery approaches such as various clustering methods can be performed in order to reveal multi-domain molecular classes. Clustering can be performed on a matrix representing the fused domains, such as for joint singular value decomposition (jSVD) [6], or can be directly computed using the DF method, such as for SNF [8]. At this point, the performance of the algorithm in terms of the clinically relevant enrichment of pathogenic classes in discrete clusters can be evaluated. For this purpose, the accuracy and meaning of the classification can be assessed using the adjusted Rand index (ARI), survival curves, or cluster quality scores (silhouette score) [10,11,12].

In our previous work, we developed and adapted data fusion approaches for the prognostic classification of uveal melanoma (UM) [6]. We performed similarity network fusion (SNF) and joint singular value decomposition (jSVD) processes, known in chemometrics as a simultaneous component analysis or simultaneous principal component analysis (PCA). We also developed the joint constrained matrix factorization (jCMF) approach based on coupled matrix factorization, also known as the k-table method, with a generalization of this factorization by allowing different constraints on the factor matrices [6]. Here, we apply DF approaches, based on open source code, for the analysis of a multi-domain cancer dataset. We apply methods based on or evolved from SNF, as developed in already published R packages, which are proven to be suited for cancer subtyping over other methods [8,9,11,13], as well as a free code version of jSVD, as previously developed in MATLAB [6]. We report on the development of the jSVD algorithm in open source code based on Python and show the details of this method. We compare the performance of this method with existing R packages [9,13] using the Skin Cutaneous Melanoma (SKCM) dataset [14].

Skin melanoma is a neoplasm of the skin, originating from melanocytic nevi, which is able to evolve into aggressive metastasizing tumors, some of which are resistant to therapy [15]. Early mutations usually occur in the genes BRAF, NRAS, or NF1, causing the constitutive activation of the mitogen-activated (MAP) kinase signaling pathway. An early period of proliferation is normally controlled by the action of the tumor suppressor gene CDKN2A. Its inactivating mutations or deletion will stimulate tumor progression [15,16]. Additional somatic mutations contribute to tumor development. The order of the mutation events varies and germline variants can contribute to disease development [17].

In 2015, 333 primary and metastatic cutaneous melanoma samples from 331 patients were analyzed by Akbani et al. [14], who considered various genomic domains, including gene expression as analyzed via RNA sequencing, somatic mutations analyzed via whole-exome sequencing, DNA methylation, and copy number alterations. All of these data were integrated and 4 different classes of risk were defined: mutant BRAF, RAS, NF1, and triple-WT (e.g., no hot spot mutations in any of the three genes) [14]. The loss of NF1 impairs the negative feedback process upon RAS activation, resulting in the activation of this gene [18]. DNA methylation has been associated with an invasive phenotype and low survival, and has a role in tumor evolution [14,19,20]. In this work, we test whether the integration of the domains of DNA methylation and gene expression can produce clinically relevant molecular classes and we test the feature extraction method for the mining of the biological meaning of such classes.

## 2. Materials and Methods

### 2.1. Data Preprocessing and Analysis

The TCGA methylation data were downloaded from Broad GDAC Firehose (https://gdac.broadinstitute.org, accessed on 1 July 2022), while the RNA-seq data were retrieved using the R package RTCGAToolbox (version 2.22.1) [21]. Only samples with methylation and RNA-seq data were considered; we performed the feature reduction process by selecting the genes with the highest MAD scores, including 1500 features for RNA-seq and the 1% most variable methylation probes, as described in [14], with matrix sizes of 332×1500 and 332×3048, respectively. The SKCM TCGA dataset was used to test the SNF process (SNFtool, version 2.3.1) [8] and two state-of-the-art data fusion methods derived from the SNF process, namely NEMO (version 0.1.0) and Spectrum (version 1.1) [9,13], which are available as R packages, as well as the jSVD method developed by our group with the Python library pymanopt (version 0.2.5) [22], based on previous studies [23]. To define the optimal number of clusters (k) and classify the patients from the U matrix produced by the jSVD function, we used the R package ConsensusClusterPlus (version 1.56.0) [24] and the implemented k-means method (complete agglomeration, Euclidean distance). The clustering and optimal number of clusters (k) were automatically computed using the network-based methods (NEMO and Spectrum). Therefore, we directly considered the classification results produced by these methods. For SNF, we set this to 3.

### 2.2. RNA-Seq Data Preprocessing

The RNA raw count data were downloaded from TCGA with the RTCGAtoolbox R package [21], organized in a matrix with genes as rows and patients as columns. All genes with a total read count lower than 100 and greater than 10^6^ were considered as outliers and removed. We kept only genes with a number of zeros lower than 20% of all patients, considering a zero count as a missing value; a similar approach had been previously applied for data fusion preprocessing [10,11]. After data filtering, we used blind vst normalization, as implemented in the DESeq 2 R package (version 1.32.0) [25], to normalize the expression data.

### 2.3. Joint Singular Value Decomposition

The joint singular value decomposition function, as previously described in [6,23], was developed in Python using the package Pymanopt [22]. The jSVD function factorizes each genomic data matrix *A* as (1):(1)$$A\approx U\Sigma_{i}V_{i}^{T}$$
where $\Sigma_{i}$ is a singular value diagonal matrix, and the other two matrices are orthonormal. The *U* matrix is used for patient clustering at the end, as it contains the fused information from all data sets. A Riemannian trust scheme was used to obtain a minimum based on the product of Stiefel manifolds (set as Product ([Stiefel (I,k), Stiefel (N1,k), Stiefel (N2,k)]), with N1 and N2 as the number of features of the RNA-seq methylation matrix). The minimization stopped when the norm of the projected gradient was lower than $1^{-12}$ (mingradnorm = 1 × 10^−12^).

### 2.4. Performance Evaluation

We used two indices to evaluate the DF methods’ performances, the adjusted Rand index (*ARI*) and silhouette score.

The Rand index (*RI*) is a measure of the agreement between two partitions of one group of elements; if we think of a set of patients classified into a set of groups by two methods, it is possible to define the *RI* as:(2)$$RI=\frac{\left(a+b\right)}{\left(a+b+c+d\right)}$$

With *a*, *b*, respectively representing the number of elements that are classified in the same group or different groups by both methods. The other values (*c*,*d*) represent elements not in agreement, such as elements in the same or different groups in only one classification method.

The adjusted Rand Index takes into account random labeling with a permutation model and is computed as:(3)$$ARI=\frac{\left(RI-E\left(RI\right)\right)}{\left(Max\left(RI\right)-RI\right)}$$

Hence, a value of 0 is defined as random labeling and 1 total agreement [26]. The *ARI* score was computed with the adj.rand.index of the pdfCluster R package (version 1.0-3) [27].

The silhouette score (*SI*) is a measure of the fit of the clustering. If we define the average distance of a point $X_{i}$ between all other members of its cluster A as *a*(*i*) and *b*(*i*) as the minimum of the distances between $X_{i}$ and all elements not in A, the *SI* for each observation can be computed as:(4)$$SI\left(i\right)=\frac{b\left(i\right)-a\left(i\right)}{max\left(a\left(i\right),\;b\left(i\right)\right)}$$

The average value of the silhouette score for each cluster (e.g., the *SI*) was computed with the silhouette function of the cluster R package (version 2.1.2) [28].

### 2.5. Feature Analysis

The significance analysis of microarray bootstrapping method as implemented in R (Samr) [29] was used to detect differentially expressed and methylated features among the jSVD derived classes in the filtered genomic domains. We evaluated the enrichment of significant features for GO biological process terms with the clusterProfiler R package (version 4.0.5) [30,31].

### 2.6. Skin Cutaneous Melanoma Molecular and Clinical Features

The molecular and clinical features of the samples used for the performance evaluation were extracted from Supplementary Table S1D, previously published in [14]. The mutation, methylation, and RNA-seq sample clusters detected by [14] in a single-domain analysis were used as a reference to compute the *ARI*. The survival curves were computed for patients using the survival data (*n* = 282) from Supplementary Table S1D of [14].

## 3. Results

### 3.1. Data Fusion

All data fusion methods were able to integrate the information based on the RNA-seq and methylation skin melanoma dataset, while maintaining some degree of similarity between single domain classes, as reported in previously published Supplementary Table S1D [14] (Table 1). As a measure of goodness of the DF process, we considered the ARI between single-domain and multi-domain clustering, where the idea is that a good data fusion model should integrate the information from the two domains and not simply rely on one of the two available. To better explain this point, we performed single-domain clustering with Spectrum [9] and later data fusion with the same method. When Spectrum was applied to a single domain (RNA or methylation), we obtained high ARI values related to the target domain clusters as compared to the others (rows one and two of Table 1). In particular, the clustering based on the methylation domain only yielded an ARI of 0.06 related to the RNA-seq clusters, as compared to a value of 0.51 for the methylation clusters. All methods were able to integrate the information in both data domains, retaining some level of cluster overlap with the single domains, but in no case did we observe a minimal level of overlap between mutation subtype classes and data fusion classes. This was expected, since the mutation classes had a low level of agreement with the single domains (Table 1, last row). Particular differences in BRAF, NF1, and KRAS variants were not observed in the three jSVD classes (Appendix A). When considering the cluster structure, the Spectrum, SNF, and NEMO techniques obtained lower silhouette scores compared to the jSVD technique. In particular, NEMO split the dataset into 12 subgroups, all of which had low silhouette scores (Table 1, last column). This behavior was related to the automatic k selection of the method, as already reported [11].

### 3.2. Feature Extraction

To observe the degree of agreement between the jSVD clusters in the single domains, we computed the differentially expressed genes and methylated probes based on the domains used for data fusion using the SAMr package [29]. The heatmap of the differentially expressed features in the RNA-seq (Appendix A) and methylation data (Figure 1) shows that clusters 1 and 3 had most of their genes downregulated in the upper part of the heatmap and their probes were hypermethylated in the lower part, while the opposite was observed for cluster 2. With regard to the TCGA annotation, clusters 1 and 3 were enriched with patients classified into the keratin, MITF-low, hyper, and CpG island methylated groups, while cluster 2 was more strongly enriched for immune class and normal class methylation elements.

The overexpressed genes in cluster 2 are enriched for GO terms associated with immune genes terms (Figure 2). If we consider the mean methylation gene levels of these genes in cluster 2 compared to 1 and 3, we see that most of them are demethylated (Figure 3). The immune gene expression subtype has been found to be associated with improved survival in different datasets and melanoma classification systems [32].

### 3.3. Survival

To verify whether the predicted clusters were associated with different survival curves, we plotted their relative Kaplan–Meier curves [33,34,35]. The class 2 patients had a better survival rate compared to class 1 and 3 patients; this class is enriched in immune RNA classes and normal-like methylation subtypes (Figure 1), which were associated with better survival in the study by Akbani et al. [14]. Cluster 1 is enriched in the CpG island methylated subtype, which has been associated with the worst survival rate [14]. Cluster 1 had the lowest survival rate (Figure 4). Class 3 is enriched with hypermethylated and hypomethylated classes that were associated with worse prognosis compared to normal-like classes [14], which are abundant in class 2 (Figure 2). The log-rank test of the survival curves showed that class 2 had a significantly better survival rate compared to classes 1 and 3. The difference between the latter two was not significant (Appendix A).

## 4. Discussion

The application of DF approaches based on the SKCM dataset reported here defined 3 clusters of patients with most of the methods presented. Based on the jSVD classification, we extracted the discriminative features of the RNA-seq and methylation domain data.

In our previous work [6], we applied DF techniques on the full UM dataset, but we rejected this approach in the case of the SKCM dataset, since it produced no meaningful results at all. This was probably related to the fact that unlike SKCM, in UM a few major genomic events, namely chromosome 3 loss and chr8q gain, are able to predict the likelihood of a patient developing metastases and the related survival rate [36]. Moreover, these major genomic events have a definite impact on RNA expression and DNA methylation [37].

Recent work shed light on the role of different genomic domains in data integration and cancer subtyping. Different combinations of miRNA, CNA, mRNA, and DNA methylation effectively depend on each other in the TCGA datasets [11], since DNA methylation generally, but not always, leads to the repression of gene expression and a copy number gain most often determines the overexpression of genes.

When evaluating the reliability of data fusion methods, the overlap between defined clusters and clinically relevant features has been assessed using scores such as ARI and log-rank test values based on survival rates and measures of cluster quality such as the silhouette score [6,11,12]. A set of mathematical indices for the evaluation of machine learning predictors in biology has been defined using bioinformatic approaches as well as the Critical Assessment of Genome Interpretation (CAGI) [38,39]. The CAGI was an experiment in which different research groups had the task to predict the phenotypes of genetic variants, such as the pathogenic effect of a missense variant on protein function [40]. In this case, the assessment was focused on detecting the distance between the real and the predicted effect, thereby evaluating the statistical significance of the results [41,42]. The same concept is difficult to apply to data fusion, since there is no gold standard reference. The assessments have so far focused on the evaluation of clusters with regard to survival and the definition of clinically relevant classes [11]. In this work, we added a single domain assessment to evaluate the effective integration of the information for distinct genomic domains. We propose a single-domain assessment as a measure of quality of DF methods.

Another general limitation of DF development is the absence of a test set, where a gold standard for machine learning is used to tune the model parameters based on a set of samples (training set) and evaluate the performance based on a different set (test or validation set) [39]. This is a general rule that is followed to avoid overfitting, a situation in which the model captures all of the variation inside the data but is not able to make any relevant predictions based on the different datasets [39]. DF methods such as SNF and jSVD can be considered unsupervised methods; therefore, they do not require parameter training. However, a different dataset to evaluate the stability in terms of the predicted clusters and the extracted features could be beneficial. The quality of features selected by means of DF could in principle be analyzed with a single plot, in which the state of the features selected from the different domains is reported. In this work, we simply checked the mean gene methylation state of overexpressed features in the RNA-seq domain (Figure 2). This approach, which relies on the general association of methylation with gene repression, has already been used to integrate the results of single-domain analyses [43,44]. Potentially, this kind of integration method combined with data fusion could support the analysis of low quality data from one genomic domain by adding information from a more reliable one. It would, for example, be possible to consider only those differentially expressed genes that are also differentially methylated or associated with a CNA event [45].

The application of data fusion in the approach shown here produced a classification result with one cluster that was very similar to the single-domain RNA-seq immuno subtype. This subtype has been associated with better survival in multiple datasets and gene expression classification systems [32]. In particular, we observed that CD19 and CD8A, which are respectively markers of B-cells and cytotoxic T-cells, were overexpressed in cluster 2. The high expression of these genes has been detected in the subgroups with high expression of immune-related genes that were associated with better prognosis [32]. Here, we show that the data fusion methods that we applied did not improve the single-domain patient classification process, at least in its unsupervised application. We cannot exclude that other DF methods that are yet to be developed could yield superior cancer classification results, and other cancers might benefit from multi-domain analyses. The fact that the three classes detected via gene expression are also visible using DF techniques is likely related to their robustness. In terms of survival, the important feature appears to be the recognition of the tumor cells by the immune system, although these tumors predate the introduction of therapy with immune checkpoint blockers [46].

Previous studies reported a potential improvement in prognostic tumor classification with data fusion compared to state-of-the-art single-domain-based analysis techniques [8]. The SNF technique was able to achieve better performance in terms of risk prediction compared to PAM50 in a breast cancer multi-domain genomic dataset [8]. The good performance of the SNF and other data fusion methods in metastatic risk prediction has been observed in UM multi-domain genomic data integration [6]. However, in both cases, the DF performance was similar to the single-domain-based prediction results [6,8], and a limited improvement would probably not balance the computational and sequencing costs required compared to a simpler single-domain analysis approach (such as PAM50).

The integration of multiple data sources can result in more robust feature selection and might improve the detection of gene pathways that are targeted by events in different genomic domains (e.g., gene methylation, pathogenic variants, CNA, etc.) [14]. The potential applications of data fusion approaches could also be based on patient similarity networks [7]; a tumor sample analyzed for multiple domains could be co-clustered with TCGA samples from the same cancer type using DF. This type of nearest neighbor classification approach could allow known risk classes to be determined for new patients. This approach could be potentially beneficial for tumors such as CM, whose pathogenic phenotype is not completely defined by the data from a single domain, such as the profile of somatic mutations [14]. This potential application of DF would probably involve several technical complications, such as batch effects related to the sequencing technologies and sample preparation. At the same time, the current results would not justify such computationally complex and expensive analysis techniques when compared to cheaper and simpler wet lab prognostic marker detection analysis techniques (such as FISH or MLPA to detect chromosome 3 monosomy in uveal melanoma or PCR for pathogenic variant detection in cutaneous melanoma [47,48]). Further work is needed to fully exploit the potential of data fusion.

## Figures and Tables

**Figure 1 biomedicines-10-03240-f001:**
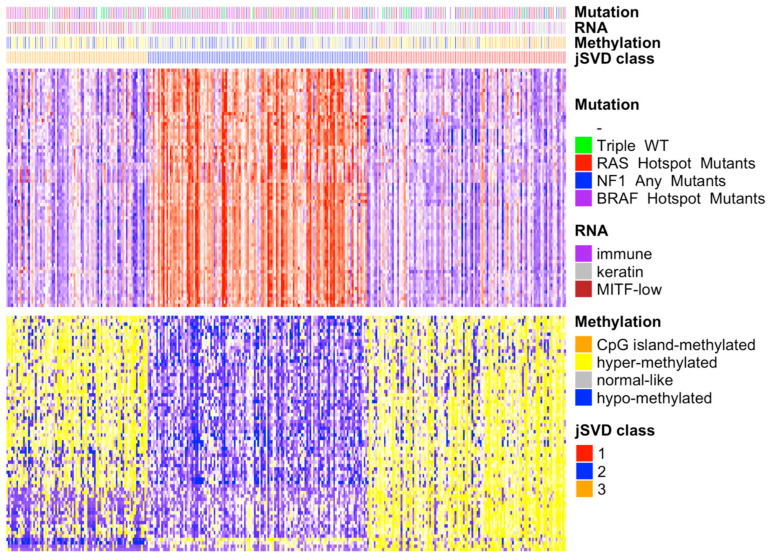
Heatmap of differentially expressed genes and methylation probes in the 3 jSVD classes. On the right is the legend of the four bars representing the TCGA SKCM somatic driver mutations, RNA and DNA methylation classes, and jSVD classes. The heatmap on top represents the RNA expression data; the gene expression is represented according to the mean values, with above mean values in red, below mean in blue, and at the mean in white; the intensity reflects the distance from the mean. The bottom heatmap shows the intensities of the DNA methylation probes; the colors range from yellow (hypermethylated, above mean) to blue (downmethylated, below mean). The bars above the heatmaps show information on the tumors obtained from the TCGA, including somatic mutation types (mutations of RAS, NF1, BRAF, or of none of these genes, triple wild-type), RNA clusters (immune, keratin, MITF-low), and methylation types (CpG island methylated, hypermethylated, normal-like, hypomethylated). Class 2 patients are enriched for immune RNA classes and normal-like methylation subtypes that were associated with better survival [14]. Cluster 1 is enriched in the CpG island methylated subtype, which was associated with the worst survival rate [14], as confirmed here (see Figure 2). Class 3 is enriched for hypermethylated and hypomethylated classes that were associated with a worse prognosis compared to normal-like classes [14], which are abundant in class 2.

**Figure 2 biomedicines-10-03240-f002:**
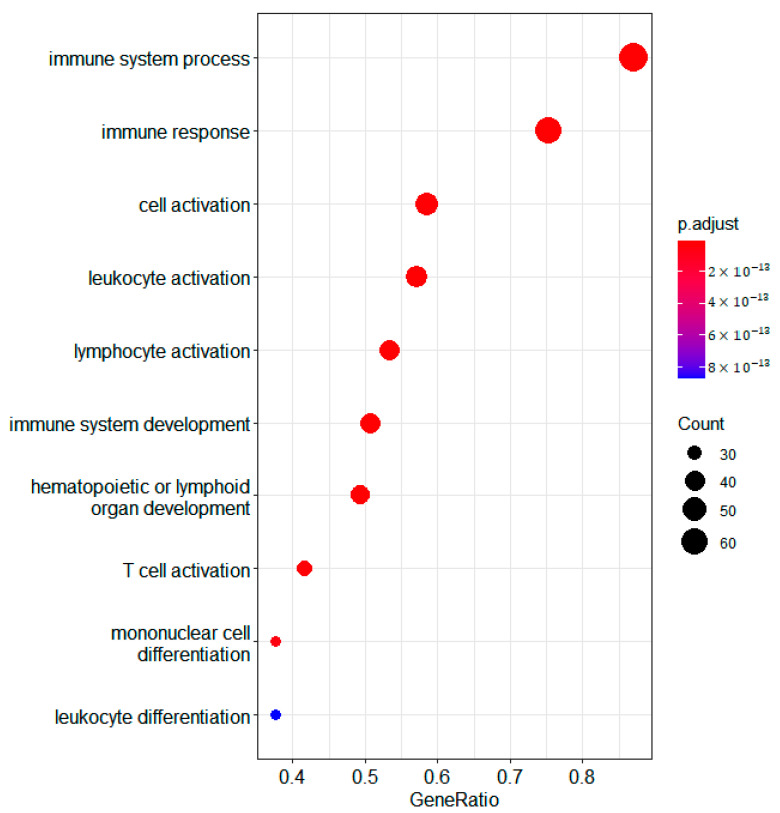
Bubble plot of differentially expressed genes in the RNA domain detected using samR. GO biological process terms for genes overexpressed in jSVD cluster 2 compared to clusters 3 and 1 are shown.

**Figure 3 biomedicines-10-03240-f003:**
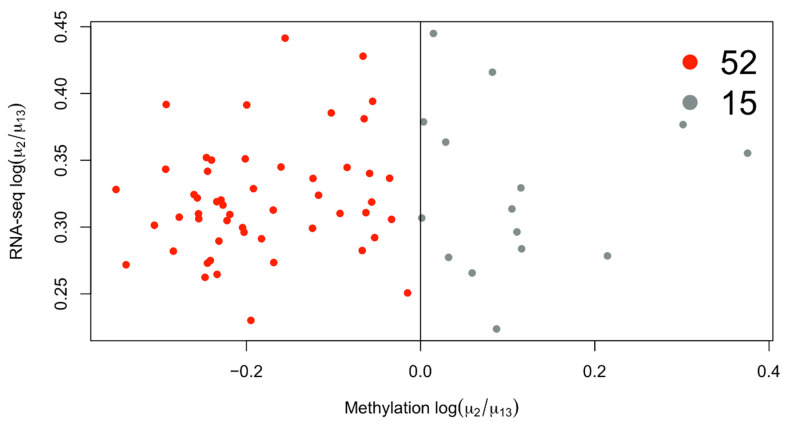
Scatterplot of differentially expressed genes in cluster 2 with available average methylation data. Most of the genes overexpressed in cluster 2 are also downmethylated, considering the average methylation level of the gene.

**Figure 4 biomedicines-10-03240-f004:**
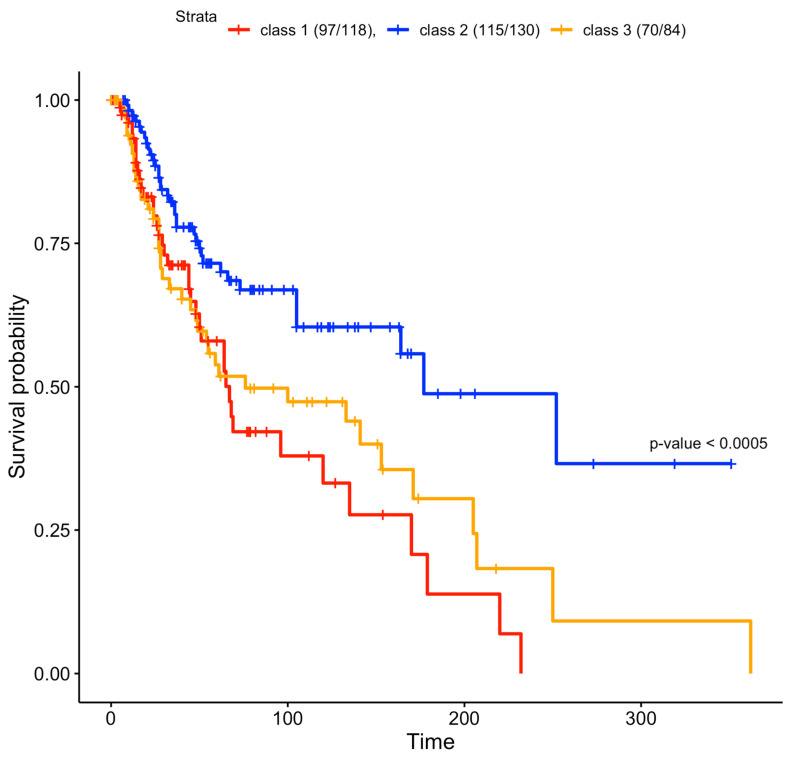
Survival curves of the three clusters found in the jSVD data fusion. Clusters 1 and 3 had worse survival rates compared to cluster 2, and were characterized by strong methylation, as shown in Figure 1. Only a subset of patients originally used for data fusion had survival data. The reported survival curves were based on 97 out of 118 class 1 patients, 115 out of 130 class 2 patients, and 70 out of 84 class 3 patients. If we exclude class 2 patients and we directly compare class 1 and class 3 curves, the log-rank test is not significant (*p* = 0.4), as in the comparison between class 2 against class 3 without class 1 (*p* = 0.1). If we compare the class 1 curve against class 2, removing class 3 patients, the log-rank test is significant (*p* = 0.0006).

**Table 1 biomedicines-10-03240-t001:** Assessment of the DF methods based on the SKCM dataset. The first three columns are the ARI scores as computed by comparing DF clusters and single-domain clusters reported in Appendix A of the TCGA paper on the SKCM dataset [14]. The fourth column reports the average silhouette score for each cluster. The first two rows report all indices computed via spectrum clustering on only RNA-seq or methylation matrices. Values of ARI close to 1 represent total agreement between two classification systems.

Genomic Domain/DF Method	Mutation Subtypes (TCGA)	RNA-seq Cluster Consenhier (TCGA)	Methylation Types 2014 08 (TCGA)	Silhouette Score
RNA-seq	−0.014	0.287	0.117	0.07, 0.135, 0.018, 0.037
Methylation	0.004	0.061	0.514	0.025, 0.026, 0.058
Spectrum	0.039	0.128	0.38	0.024, 0.03, 0.026
SNF	0.027	0.152	0.275	0.008, 0.004, 0.004
NEMO	0.029	0.108	0.195	0.05, −0.017, −0.015, −0.03, −0.008, 0.005, 0.031, −0.014, −0.027, −0.019, −0.023, −0.009
jSVD	0	0.272	0.147	0.212, 0.339, 0.281
Mutation subtypes (TCGA)	1	0.007	0.021	NA

## Data Availability

The data can be downloaded using the RTCGAToolbox, as shown in the attached scripts, or with the TCGAbiolinks R package. The data are also available on the Broad GDAC Firehose website (http://gdac.broadinstitute.org/, URL accessed on 6 December 2022). All scripts used for this work are available on the github repository (https://github.com/FranzReg91/Amaro_et_al_2022_SKCM, URL accessed on 6 December 2022).

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
