# Peer review of "Evaluation and Comparison of Multi-Omics Data Integration Methods for Subtyping of Cutaneous Melanoma"

_biomedicines, 2022, doi:10.3390/biomedicines10123240_

Round 1

Reviewer 1 Report

In this manuscript, Adriana Amaro and colleagues describe the comparative analysis of multi-omics datasets using data fusion techniques. This topic is of great interest given the rather limited experience and knowledge available. Interestingly, the authors conclude that data fusion is not superior to separate analysis at the single-omics level.

The manuscript is quite well written, the results and interpretation seem plausible, and I am in favor of publishing this manuscript. However, there are two important issues that the authors should clarify before publication.

1) The analysis strategy is not described in enough detail. In particular, I miss a reference to the dimensionality of the data and an explicit description of the ground truth that the authors use as a reference. 

2) The Github repository needs to be made available for the review process. https://github.com/FranzReg91/Amaro_et_al_2022_SKCM is not accessible at present. The code base of this manuscript is an important reference that not only helps clarify the analysis details, but also serves as a toolbox for researchers interested in implementing data fusion.

Reviewer 2 Report

The manuscript entitled "Evaluation and comparison of multi-omics data integration 2

methods for subtyping of cutaneous melanoma" carries significance and novelty for cutaneous melanoma research. It was well prepared, and the results were clearly presented. In order to obtain recommendation for acceptance for publication, the authors need to answer/address the following questions/concerns: 

1. How many patients exactly were included in their analyses? The text mentioned a number of 331 patients, but that was not really the total numbers of patients that were analyzed, correct? Those numbers are important so the authors need to clearly list them in the manuscript.

2. The overall conclusion from this study was rather negative by saying that data fusion does not generate better classification. Why was that? Patients numbers too low? Cancer type-specific? 

3. Would it require too much work if the authors perform similar analyses for a cancer type with much more patients, say CRC or breast cancer? A different cancer type may give the authors quite different conclusions, so this reviewer request that the authors look into this direction. 

Round 2

Reviewer 1 Report

The authors successfully addressed my concerns. I now fully support  publication of this manuscript.

Reviewer 2 Report

Ready to be accepted for publication.